# Challenges and Strategies for Enhancing eHealth Capacity Building Programs in African Nations

**DOI:** 10.3390/jpm13101463

**Published:** 2023-10-05

**Authors:** Flora Nah Asah, Jens Johan Kaasbøll

**Affiliations:** HISP Centre, Department of Informatics, University of Oslo, Gaustadallen 30, 0373 Oslo, Norway; jens@ifi.uio.no

**Keywords:** eHealth, ICTs, Capacity building activities, BETTEReHEALTH

## Abstract

eHealth applications play a crucial role in achieving Universal Health Coverage. (1) Background: To ensure successful integration and use, particularly in developing and low/middle-income countries (LMIC), it is vital to have skilled healthcare personnel. The purpose of this study was to describe challenges that hinder capacity-building initiatives among healthcare personnel in developing and LMIC and suggest interventions to mitigate them. (2) Methods: Adopted a descriptive research design and gathered empirical data through an online survey from 37 organizations. (3) Results: The study found that in developing and LMIC, policymakers and eHealth specialists face numerous obstacles integrating and using eHealth including limited training opportunities. These obstacles include insufficient funds, inadequate infrastructure, poor leadership, and governance, which are specific to each context. The study suggests implementing continuous in-service training, computer-based systems, and academic modules to address these challenges. Additionally, the importance of having solid and appropriate eHealth policies and committed leaders were emphasized. (4) Conclusions: These findings are consistent with previous research and highlight the need for practical interventions to enhance eHealth capacity-building in LMICs. However, it should be noted that the data was collected only from BETTEReHEALTH partners. Therefore, the results only represent their respective organizations and cannot be generalized to the larger population.

## 1. Introduction

According to a recent United Nations (UN) report, almost half of the global population lacks access to essential healthcare services [1]. This is primarily due to over 800 million individuals allocating less than 10% of their household budget toward health expenses. As a result, millions worldwide are facing significant challenges and cannot afford necessary healthcare services [2]. The World Health Organization (WHO) Director-General urged leaders and policymakers, particularly those from developing countries, to embrace eHealth to improve healthcare. By integrating digital health services, access to and quality of health service delivery can be enhanced, ultimately contributing to achieving the United Nations Sustainable Development Goals [3,4]. This manuscript is an adapted version of a paper originally published in the IOS SHTI pHealth 2022 Proceedings [5].

eHealth refers to the provision of health services and information through the Internet, utilizing digital technologies such as information communication technologies (ICTs) and data to offer healthcare services [6,7]. These services may include physical and psychological diagnosis and treatment, telepathology, vital signs monitoring, electronic prescribing, and teleconsultation [8]. The literature has extensively documented the benefits of integrating eHealth in healthcare, such as improving healthcare delivery, efficacy, and quality of care [9]. eHealth applications are considered a potential “game changer” that could improve access to affordable and effective healthcare services [10,11]. Additionally, eHealth services can enhance patients’ health-related knowledge and behavior, facilitate information exchange between healthcare providers and patients, and improve coordination and continuity of care while reducing the cost of healthcare delivery. In the long term, eHealth has the potential to transform the workflow of healthcare and support the achievement of Universal Health Coverage (UHC) [11,12,13].

### Factors Influencing the Integration and Use of eHealth

Ensuring access to quality healthcare without financial barriers is essential to achieving Universal Health Coverage (UHC) [3]. eHealth technologies have been identified as key enablers of UHC. However, their integration and widespread use are limited, particularly in developing countries where they are most needed [14]. This is due to several factors, including the high cost of IT infrastructure and a need for more skilled personnel to adopt eHealth [15]. A study on integrating eHealth in Tanzania found that inadequate ICT skills, high cost of ICT, under-developed IT infrastructure, and a need for more information about appropriate ICT solutions were major hindrances [16]. While eHealth services can improve healthcare quality, factors such as insufficient budget for ICT infrastructure, security, privacy, and confidentiality concerns can hinder their integration [17]. A qualitative study on implementing a standardized information system in Cameroon found that centralized structures deter the allocation of finance to ICT equipment, particularly at lower health system levels [18]. According to Mars & Scott [19], LMIC governments need more financial resources, resulting in cautious spending on health activities. Adebesin et al. [20] also pointed out that the lack of interoperability of Health Information systems (HIS) hinders eHealth integration. A survey on eHealth adoption obstacles in Africa revealed limited participation in eHealth standards development beyond the International Organization for Standardization’s requirements. Stiawan [21] further explained that the inability of information systems to exchange and share data and information among government agencies is a significant obstacle to eHealth integration. Similarly, Sluijs et al. [22] noted that the need for standards prevents government institutions, such as hospitals, from achieving their targets. The data and information needed by health personnel, such as population data, health insurance data, and patients’ medical records, are often stored on different systems and managed by various government departments, making interoperability crucial. S. Masud et al. [23] emphasized standardizing data and information formats to achieve interoperability.

Issues with eHealth policies [24] and leadership [25] within the public sector have been identified as areas of concern in LMICs. Poor coordination among government departments and inadequate policies are challenges in integrating eHealth initiatives, according to Luna et al. [26]. In a study assessing eHealth policies in four African countries, authors noted that strategic goals were vague and lacked consolidated plans [27]. Weak leadership within the government can also hinder the coordination of eHealth projects at the national level, as noted by Mburu et al. [28]. Additionally, developing long-term strategies can be challenging in unstable political environments. However, the political will to embrace eHealth is growing in Sub-Saharan Africa, with the African Union and WHO working with LMIC governments to harmonize eHealth activities on the continent, according to Mars [19].

Various challenges impede the successful integration and use of eHealth services in the healthcare sector. These include the lack of computer equipment, poor internet connectivity, and inconsistent electricity supply [10,29]. Studies have revealed that healthcare workers’ reluctance to embrace technology significantly limits their participation in eHealth activities. This is often due to their need for sufficient knowledge and skills to operate eHealth services, which results in denial and resistance towards information technologies [30,31,32]. Furthermore, the low usage of the internet among doctors in Pakistan is attributed to their insufficient IT skills. In low- and middle-income countries, inadequate human resources pose a significant threat to the successful integration and use of eHealth [33]. The current capacity-building activities for IT professionals need improvement, as studies have shown that the need for qualified health professionals is a persistent problem. Sufficient knowledge and skills are crucial for healthcare providers to use eHealth services effectively and keep up with technical advancements in an ever-changing eHealth environment [34,35,36,37].

Research conducted in LMICs suggests that enhancing the ICT skills of health personnel through education and training is crucial [9,38]. Prior studies have primarily focused on the availability of human resources and ease of use rather than the competencies and skills of health personnel in eHealth in Africa and LMICs. While these studies are informative, they challenge policymakers in comprehending, evaluating, and addressing obstacles. Our review unveiled a scarcity of articles on capacity-building endeavors among health personnel, underscoring the importance of this study.

## 2. Objectives

This study delves into the challenges that hinder capacity-building initiatives among healthcare personnel in developing and LMIC and suggests interventions to mitigate them. The data from this study was extracted from a large study conducted within the BETTEReHEALTH project identifying challenges of integrating digital health policies and gaps in developing digital health capacity among healthcare professionals within the BETTEReHEALTH project. However, in this article, we focused on identifying gaps in eHealth capacity-building activities and suggested approaches to mitigate them. The study gathered empirical data from an online survey on capacity-building activities in Africa. The significance of building the capacity of healthcare personnel in eHealth cannot be overstated, as a well-trained workforce in this field will strengthen health systems and enhance access to and quality of healthcare delivery [39]. The results of this study will guide the provision of health-related information and resources to BETTEReHEALTH partners and others and serve as a roadmap to measure and alleviate the barriers.

BETTEReHEALTH is a European Union project and funded by European Union Horizon 2020. The project aims to strive to increase the level of international cooperation in eHealth, inform and strengthen end-user communities and policy makers in making the right decisions for the successful implementation of e-Health. The project’s purpose is to increase opportunities for stakeholders in Africa and Europe with the overall aim of better health outcomes through better healthcare accessibility and higher quality. BETTEReHEALTH provides a platform for stakeholders to network, disseminate and communicate, and provide information on best practices, lessons learnt and policy guidance on eHealth. The project has four hubs: Ghana (Western region), Malawi (Southern region), Ethiopia (Central and Eastern regions), and Tunisia serving the Northern region.

## 3. Methods

In the research on which this article is based, a descriptive design was adopted, and data was gathered through an online survey created with Google Forms. A links to the following files (i) information form explaining the purpose of the study, (ii) informed consent form, (iii) survey, were sent to project leaders of four BETTEReHEALTH hubs i.e., Ghana, Malawi, Ethiopia, and Tunisia. They then forwarded the survey to eHealth organizations/institutions in their respective countries. The online survey had four sections and five questions per section. The four sections were:eHealth capacity building activities.Factors hindering eHealth capacity building among health professionals.Health workers IT literacy.Proposed suggestions to build IT skills among health professionals.

The project leader, who was the second author, developed the questionnaire, and the BETTEReHEALTH project managers in the hubs were asked to review it. To ensure the accuracy of the questionnaire, it was pre-tested among master students from the Department of Informatics at the University of Oslo. This helped to assess if the questions were unambiguous and easily understood by the respondents and provided direct evidence of questionnaire data validity. Based on feedback from the pre-test, some questions were modified and clarified to improve their quality.

The survey questions were written in English and were open to participation for two months. The questions were closed-ended and were scored on a five-point Likert type scale ranging from strongly agree to strongly disagree. Managers in decision-making positions responded to the questionnaire. Three follow-up participation requests were sent out every two weeks. Ethical clearance for this study was granted as part of a larger BETTEReHEALTH project. In addition, all participants were informed about the study; a consent form and information sheet were attached to the questionnaire. A total of 37 organizations/institutions responded from 13 countries, with one excluded from analysis due to not indicating the name of the country. In addition, four managers were selected at random and interviewed informally to gain insight into eHealth capacity-building activities within their organizations.

### Quantitative and Qualitative Data Analyses

The responses to the survey questions were collected and exported to an Excel spreadsheet. To ensure the anonymity of respondents, all metadata was removed from the file. The data gathered was processed with SPSS statistical software. To help us in analyses, the data were arranged in the following sections including general characteristics of the respondents, capacity-building activities and gaps in capacity-building programs, factors hindering eHealth capacity-building activities, health workers IT literacy, and suggestions to building IT skills of health personnel. Thereafter, the institutions were grouped by countries and then by regional hubs. Since there were only 37 respondents, descriptive analyses were employed to summarize the results.

The interviews were analyzed using content analysis, which involves identifying and categorizing themes within text data. Comments from the online survey were also analyzed using this method. The results were supported by quotes from the interviews. However, it should be noted that the survey results only represent their respective organizations and cannot be generalized to the larger population.

## 4. Results

### 4.1. General Characteristics of Respondents i.e., Organizations

Thirty-seven organizations from 13 countries responded to the survey. The BETTEReHEALTH project has four hubs namely Ghana (Western), Malawi (Southern), Tunisia (Northern), and Ethiopia (Central and Eastern) serving four geographical regions. We divided the responses (37) per regional hub to ascertain the number of responses per hub. Most responses came from the Southern region and the least number of responses came from the Northern and Western regions. See Table 1.

After analyzing the responses, they were sorted based on the types of organizations. The findings reveal that 36% (13) were institutes of higher education, 34% (12) were government agencies, and 11% (4) were NGOs. Refer to Figure 1 for a visual representation of the respondents/organizations that participated in the study.

### 4.2. Capacity Building Activities

We evaluated capacity-building activities and discovered that organizations engage in various types of such activities. These include pre-education, in-service training, and support from external specialists, among others. Although pre-education and in-service training were the most prevalent activities, we observed that “support from specialists outside the organization” was the least employed activity, as shown in Figure 2. We asked the participants why this activity was not widely used, and one manager explained that it involves hiring a specialist, which has financial implications that most organizations cannot afford.

#### Gaps in Capacity Building Activities

Though the organizations surveyed had different capacity-building activities, our survey observed some gaps. For instance, the current capacity-building programs do not cater to IT professionals, manager/administrative health personnel, eHealth specialists, and policymakers as shown in Figure 3.

### 4.3. Factors Hindering eHealth Capacity-Building Activities

The importance of having digital skills is widely recognized, but our survey revealed that there are several challenges that hinder eHealth capacity-building efforts. In this section, the data was further analyzed according to the four geographical regions. The findings revealed that in the Northern hub, infrastructural constraint and lack of motivation were factors hindering eHealth capacity building. While lack of financial support and infrastructural constraint were the most frequent factors that hinder eHealth capacity building in the Southern, Central and Eastern hubs, as illustrated in Figure 4.

### 4.4. Health Workers’ IT Literacy

Regardless of their profession or level of digital expertise, it is widely agreed that possessing eHealth skills will greatly impact an individual’s career. IT skills were found to be the most valuable in our study. The data was also analyzed based on income levels of countries, we observed that more individuals in low- and middle-income countries have access to and use smartphones over computers. As a result, more people are becoming proficient in using smartphones, as depicted in Figure 5.

### 4.5. Proposed Suggestions to Build ICT Skills of Health Personnel

During the analysis of the interviews, the most frequent recommendations made were to offer ICT in-service training, introduce computer-based systems, and enhance Internet accessibility. Additionally, one respondent highlighted the importance of providing health workers with visual presentation skills, while another emphasized the need for more pre-and in-service training activities. For further suggestions, refer to Figure 6.

## 5. Discussion

Our research aimed to uncover the challenges faced by healthcare professionals when developing eHealth capacity-building initiatives within the BETTEReHEALTH community. Our findings revealed several factors that hinder the progress of such programs, with the lack of comprehensive eHealth capacity-building policies being a significant obstacle [20]. These policies are essential in creating a shared understanding of eHealth objectives and prioritizing associated efforts [28]. Our study also found that in low- and middle-income countries (LMICs), eHealth policies are often too broad and require more specific delineation of the roles and responsibilities of various stakeholders. For instance, inadequate eHealth policies in Ethiopia and Ghana have resulted in disparities in how the government and research communities implement mHealth activities [40]. For example, mHealth activities are uncoordinated and do not align with national health priorities. As Khoja et al. [41] highlighted, policymakers must take a proactive approach in developing policies that enable seamless and reliable planning of eHealth programs.

Our survey revealed that eHealth training options are diverse, but limited for policymakers, IT professionals, and managers. This supports the findings of a previous study by [35], which highlighted the importance of continuous and practical capacity building for IT specialists and professionals. Our recent online survey also uncovered various obstacles that hinder capacity-building activities, ranging from systemic issues like insufficient infrastructure, low budgets, weak government policies, and poor governance at the national level, to individual barriers such as lack of time, skills, and motivation. Our findings are in line with other researchers who have emphasized the critical role of national eHealth policies in bridging gaps in eHealth activities [19,20,33,42,43]. It’s worth noting that while adequate financing is crucial for infrastructure development, technological tools, training workshops, and qualified personnel, it can only be utilized effectively and efficiently with strong policies, political commitment, and good leadership [36].

Our survey has clearly indicated that individuals living in rural areas who have access to smartphones and other technologies tend to develop more IT skills in comparison to those who use computers and keyboards. Our findings support previous studies that highlight the importance of high mobile service penetration in aiding healthcare efforts in areas with limited resources [44]. We emphasize that the skills and knowledge of health personnel are vital, as eHealth tools cannot be effectively utilized without them.

The American Medical Informatics Association advocates for a system-wide approach that integrates digital skills training early in students’ education as a compulsory aspect of existing school programs to address skill gaps. As a result, we established a partnership with five universities—Eduardo Mondlane in Mozambique, University of the Western Cape in South Africa, University of Dar Es Salaam in Tanzania, University of Malawi, and University of Gondar in Ethiopia—for the DEDICATED project (BETTEReHEALTH project). We have designed ten eHealth modules that will be taught to undergraduate and postgraduate students at these universities, with the aim of building the capacity of future eHealth professionals [45]. The DEDICATED initiative is in its infancy, but the concept is used at other institutions. The European Health Parliament, for instance, has recommended establishing mandatory customized training programs on digital skills for health professionals. This training should start from the early education phase and extend to professional development programs [45]. Similarly, RAFT uses the same approach to train medical doctors in 15 francophone central and West African countries [19] and capacity building through professional bodies and societies—for example, edX for business. edX is a learning management system (LMS) offering learning solutions that align with the growth objectives of every staff member within an organization. LMS platform offers a simple and efficient set-up process with advanced real-time learning opportunities. Our research has shown that local capacity-building initiatives are effective strategies to develop skilled healthcare staff [9]. In addition, we strongly recommend that educational curricula for health personnel should also include eHealth skills training. While training is essential, it can only succeed with solid eHealth policies and committed leaders.

## 6. Conclusions

The healthcare industry is characterized by constant evolution, necessitating professionals who can adapt to its changing demands. Nonetheless, for eHealth education and training services to be fully effective, it is crucial to have adequate funding, infrastructure, leadership, governance, and qualified human resources at all levels of the health system. Despite the potential advantages of integrating eHealth services, many low- and middle-income countries (LMICs) face numerous obstacles. Our research underscores the importance of providing healthcare professionals with eHealth skills that are context-specific and tailored to diverse groups. By addressing the challenges facing eHealth capacity building, we can improve health service delivery and contribute to realizing universal health coverage (UHC). Our study offers valuable insights into eHealth capacity building and innovation promotion initiatives for public health and healthcare professionals, adding to the ongoing conversation on promoting innovation and building eHealth capacity. While the data is based on a relatively small sample of 37 respondents from 15 African countries, the findings raise broader issues relevant to implementing eHealth in resource-constrained settings. It is important to note that the study had some limitations, but we took steps to enhance its validity. We ensured that the data collection method was appropriate, and that the questionnaire was clear, concise, and reviewed by project managers. Additionally, we pilot-tested the questionnaire to avoid ambiguity. The survey respondents were managers, and their in-depth knowledge of the subject lends weight to our results. Furthermore, we collected the data systematically and rigorously, enhancing the study’s validity and reliability. It is worth noting that the results only reflect the perspective of the organizations and cannot be applied to the larger population. While future studies with a larger sample size would yield more comprehensive results, our findings are consistent with previous research.

## Figures and Tables

**Figure 1 jpm-13-01463-f001:**
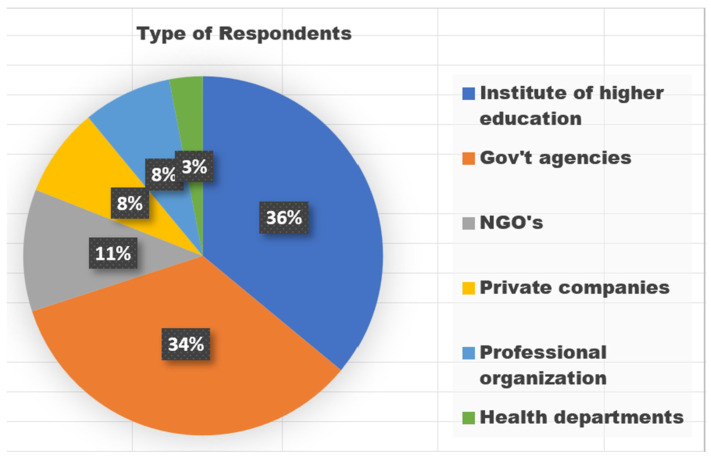
Type of Respondents who participated.

**Figure 2 jpm-13-01463-f002:**
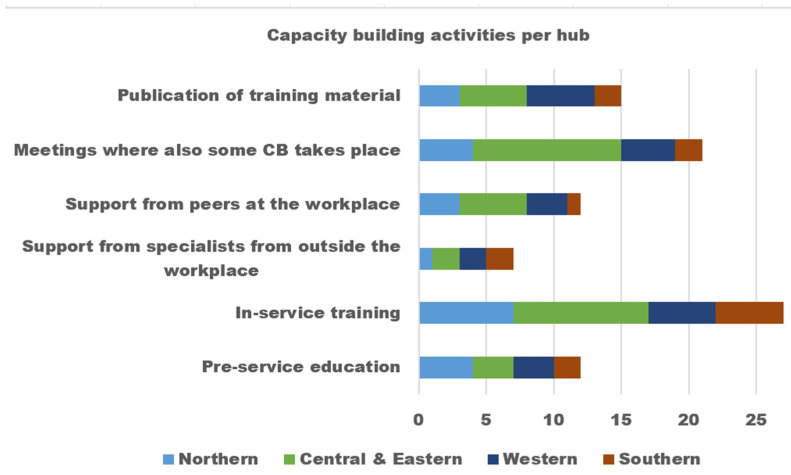
eHealth capacity building activities by hubs.

**Figure 3 jpm-13-01463-f003:**
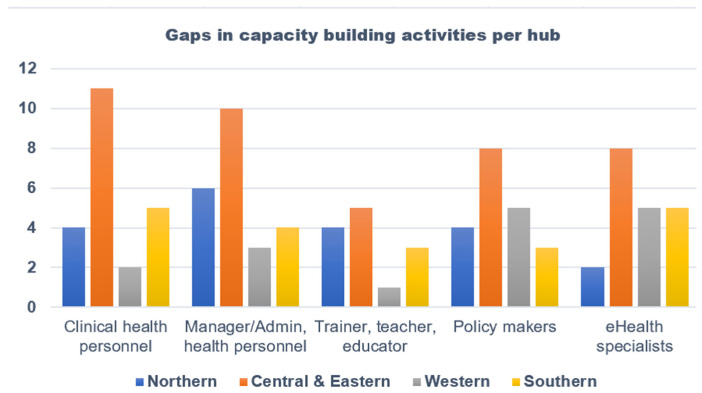
Professional groups targeted for capacity-building activities per hub.

**Figure 4 jpm-13-01463-f004:**
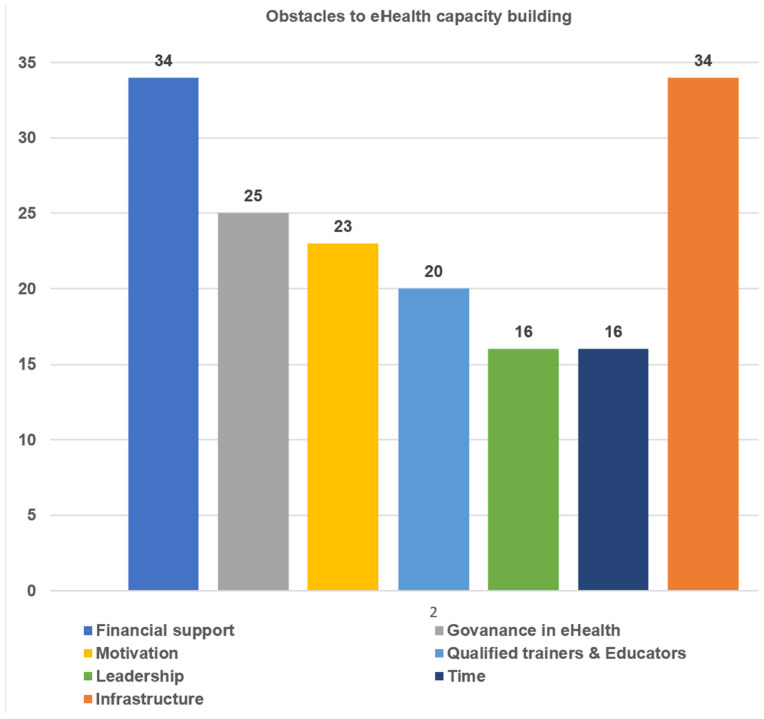
Obstacles to capacity building activities.

**Figure 5 jpm-13-01463-f005:**
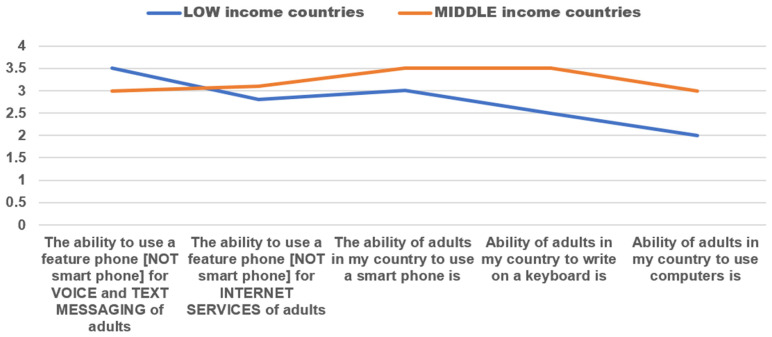
ICT level of the General Public. 1 = very low, 6 = very high.

**Figure 6 jpm-13-01463-f006:**
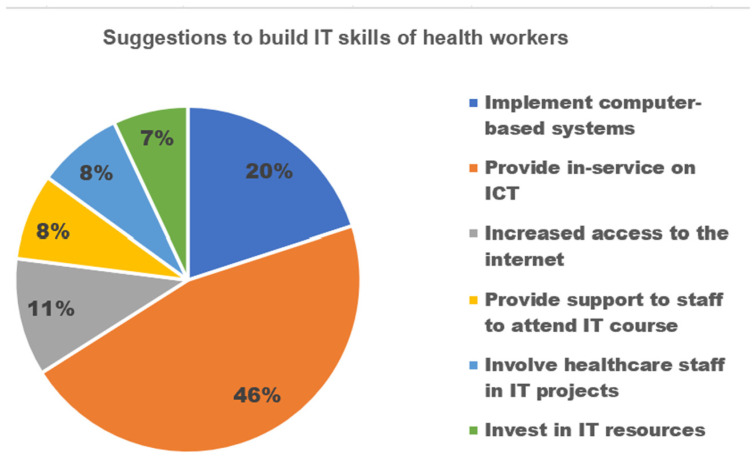
Suggestion to build IT skills of health workers.

**Table 1 jpm-13-01463-t001:** Description of Respondents.

Names of Country	No. of Responses	Regions
Malawi	6	Southern
Tanzania	2	
South Africa	2	
Mozambique	4	
Ethiopia	5	Central & Eastern
Kenya	3	
Uganda	1	
Ghana	4	Western
Togo	3	
Tunisia	3	Northern
Mauritania	2	
Morocco	1	
Algeria	1	

## Data Availability

The datasets used and analyzed during the current study are available from the corresponding authors on reasonable request.

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
