# Peer review of "Challenges and Strategies for Enhancing eHealth Capacity Building Programs in African Nations"

_jpm, 2023, doi:10.3390/jpm13101463_

Round 1
Reviewer 1 Report
This is an interesting paper about the capacity-building initiatives among healthcare personnel in developing countries.
Some suggestions:
In Methods section please give more details about the questionnaire. How many questions/items does it has? What factors does it measure? Is it self-developed? Did you tested/use it before? What about the validity and the reliability? Also, please give more details about the items. Open questions? Multiple choice? Likerd?
For data analyses, what about some hypotheses tests? chi-sqare tests? In you results I can read some comparisons. Please describe how did you perform these in advance (methods section). I strongly suggest to perform some statistic test to examine the significance of the differences/comparisons.
Major revision is required.
Author Response
Dear Reviewer,
Thank you for giving me the opportunity to submit a revised draft of my manuscript. We appreciate the time and effort you have dedicated to providing valuable feedback on our manuscript. We are grateful to the reviewers for their insightful comments on our paper. We have been able to incorporate changes to reflect most of the suggestions provided. We have highlighted the changes within the manuscript. Here is a point-by-point response to the reviewers’ comments and concerns
Comment 1. In Methods section please give more details about the questionnaire. How many questions/items does it has? What factors does it measure? Is it self-developed? Did you tested/use it before? What about the validity and the reliability? Also, please give more details about the items. Open questions? Multiple choice? Likerd?
Response: Thank you for pointing this out. We have accordingly made some revisions to include the points raised. You can find the explanation in the "Method," 3rd paragraph.
Comment 2. For data analyses, what about some hypotheses tests? chi-sqare tests? In you results I can read some comparisons. Please describe how did you perform these in advance (methods section). I strongly suggest to perform some statistic test to examine the significance of the differences/comparisons.
Response: Thank you for this suggestion. It would have been
interesting to explore this aspect. However, in the case of our study, it seems slightly out as we did not use hypothesis tests, as the purpose of the study was to describe a fact, phenomenon, or situation. We were merely trying to bring these facts together and to describe the situation rather than experimenting. As already indicated, the purpose of this paper was to describe challenges that hinder capacity-building initiatives among healthcare personnel in developing and LMIC and suggest interventions to mitigate them. The responses received were small; 37 countries participated. 37 is a small number to perform statistical tests.
Reviewer 2 Report
1. Clarify research objectives in the abstract: The abstract should explicitly state the specific research objectives to provide a clear overview of the study's purpose and goals.
2. Provide more methodological details: Expand on the methodology used in the online survey, including information about survey design, participant selection criteria, and data collection process to enhance the study's rigor and replicability.
3. Consider a larger sample size: Given the significance of the study's findings, it is advisable to increase the sample size to improve statistical reliability and to generalize the conclusions to a broader population.
4. Define BETTEReHEALTH: Offer a concise definition or background information about BETTEReHEALTH to contextualize its relevance and significance in the study.
5. Elaborate on context-specific challenges: Provide further insights into the context-specific obstacles identified, such as insufficient funds and inadequate infrastructure, by explaining how these challenges manifest in the context of low and middle-income countries in Africa.
6. Include study limitations: Acknowledge potential biases or constraints in data collection and analysis, and discuss the study's limitations to provide a balanced perspective on the research findings.
7. Support recommendations with evidence: Strengthen the proposed strategies for enhancing eHealth capacity building by citing relevant evidence or existing literature to demonstrate their effectiveness and feasibility.
8. Address data collection bias: Given that data was collected only from BETTEReHEALTH partners, discuss the potential for bias and outline efforts made to mitigate bias or plans for broader data collection in the future.
9. Highlight practical implementation of strategies: Elaborate on the practical aspects of implementing the proposed plans, such as estimated costs, timelines, and potential challenges in their execution.
10. Compare with previous research for novelty: Emphasize the study's originality and contribution by comparing its findings with previous research on eHealth capacity building in African nations, highlighting how it adds new insights to the existing knowledge base.
11. All the graphical plots represented here are of poor quality. Needs appropriate potting by using relevant tools suitable for journal standards.
Overall, the manuscript needs major revisions before further processing.
The manuscript needs minor moderate English and language revisions before further processing.
Author Response
Dear Reviewer,
Thank you for giving me the opportunity to submit a revised draft of my manuscript. We appreciate the time and effort you have dedicated to providing valuable feedback on our manuscript. We have been able to incorporate changes to reflect most of the suggestions provided. We have highlighted the changes within the manuscript.
Here is a point-by-point response to the reviewers’ comments and concerns.
Comment 1: Clarify research objectives in the abstract: The abstract should explicitly state the specific research objectives to provide a clear overview of the study's purpose and goals.
Response: Agree. We have, accordingly, modified the abstract to emphasize this point.
Comment 2: Provide more methodological details: Expand on the methodology used in the online survey, including information about survey design, participant selection criteria, and data collection process to enhance the study's rigor and replicability.
Response: Agree. We have, accordingly, modified the methods section to emphasize this point.
Comment 3: Consider a larger sample size: Given the significance of the study's findings, it is advisable to increase the sample size to improve statistical reliability and to generalize the conclusions to a broader population.
Response: Thank you for this suggestion. It would have been
interesting if we had a larger sample size. At this moment, we cannot change the sample.
Comment 4: Define BETTEReHEALTH: Offer a concise definition or background information about BETTEReHEALTH to contextualize its relevance and significance in the study.
Response: Agree. We have, accordingly, modified the objectives to emphasize this point. Refer to the section of Objectives, 2nd paragraph.
Comment 5: Elaborate on context-specific challenges: Provide further insights into the context-specific obstacles identified, such as insufficient funds and inadequate infrastructure, by explaining how these challenges manifest in the context of low and middle-income countries in Africa.
Response: Thanks for your suggestions. However, the purpose of the study is to describe the factors hindering eHealth capacity-building activities.
Comment 6: Include study limitations: Acknowledge potential biases or constraints in data collection and analysis, and discuss the study's limitations to provide a balanced perspective on the research findings.
Response: Agree. We have, accordingly, modified the conclusion section to emphasize this point. You can find the modifications in paragraphs 2 and 3 in the conclusion.
Comment 7: Support recommendations with evidence: Strengthen the proposed strategies for enhancing eHealth capacity building by citing relevant evidence or existing literature to demonstrate their effectiveness and feasibility.
Response: We did not quite understand this comment.
Comment 8: Address data collection bias: Given that data was collected only from BETTEReHEALTH partners, discuss the potential for bias and outline efforts made to mitigate bias or plans for broader data collection in the future.
Response: Agree. We have, accordingly, modified the conclusion section to emphasize this point. You can find the modifications in paragraphs 2 and 3 in the conclusion
Comment 9: Highlight practical implementation of strategies: Elaborate on the practical aspects of implementing the proposed plans, such as estimated costs, timelines, and potential challenges in their execution.
Response: Thanks for the suggestions. However, this is outside the objective of this study.
Comment 10: Compare with previous research for novelty: Emphasize the study's originality and contribution by comparing its findings with previous research on eHealth capacity building in African nations, highlighting how it adds new insights to the existing knowledge base.
Response: Thanks for the suggestion. While this study is not novel, the findings offer valuable insights into eHealth capacity-building and innovation promotion initiatives for public health and healthcare professionals and contribute to the conversation on promoting innovation and building eHealth capacity among public health and healthcare professionals.
Comment 11: All the graphical plots represented here are of poor quality. Needs appropriate potting by using relevant tools suitable for journal standards.
Response: We did not quite understand this comment.
Reviewer 3 Report
The study is important to readers because it shows the challenges and strategies to improve eHealth capacity development programs. Applied to a few African nations.
The study proves that the lack of trained health care personnel and the low income of these countries impede their development.
The sample is considered small with only 37 BETTEReHEALTH partner organizations from 15 countries.
Authors should expand:
1. About the obstacles according to the context found (insufficient funds, inadequate infrastructure, leadership and governance).
2. Topics of greatest deficiency for continuous training
3. How can leadership be evaluated?
4. What are the practical and simple interventions that you propose?
5. Update the following references:
Ref 2: from 2005
Ref 10 and 17: from 2007
Ref 20, 36 and 43: from 2010
Ref 23 and 25: from 2006
Line 48, 86: Space between text and reference.
Explain Figure 1, 2 before text. First explain Figure xxx, and after show the Figure.
Author Response
Dear Reviewer,
Thank you for giving me the opportunity to submit a revised draft of my manuscript. We appreciate the time and effort you have dedicated to providing valuable feedback on our manuscript. We are grateful for their insightful comments provided on our paper. We have been able to incorporate changes to reflect most of the suggestions provided. We have highlighted the changes within the manuscript.
Here is a point-by-point response to the reviewers’ comments and concerns.
Comment 1: About the obstacles according to the context found (insufficient funds, inadequate infrastructure, leadership and governance).
Response: We did not quite understand this comment.
Comment 2: Topics of greatest deficiency for continuous training.
Response: We did not quite understand this comment.
Comment 3: How can leadership be evaluated?
Response: Thanks for this suggestion. However, this is a descriptive study. Evaluating leadership was not part of the objective of this study.
Comment 4: What are the practical and simple interventions that you propose?
Response: Thanks for your comment. These aspects have been discussed extensively in the conclusion. Read the 3rd paragraph in the Conclusion.
In addition to the above comments, all spelling and grammatical errors pointed out by the reviewers have been corrected.
We look forward to hearing from you.
Kind regards,
Flora
Round 2
Reviewer 2 Report
Support recommendations with evidence: Strengthen the proposed strategies for enhancing eHealth capacity building by citing relevant evidence or existing literature to demonstrate their effectiveness and feasibility.
The above comment suggested authors to compare the outcomes with previous studies from literature. To support the outcomes and directions in present study.
Also, The Piecharts should be on white background and should be organized well. The plots are not satisfactory. Kindly revise
Needs moderate language editing
Author Response
Dear Reviewer,
Thank you for giving us the opportunity to submit a revised draft of our manuscript. We appreciate the time and effort you have dedicated to providing valuable feedback on our manuscript. We are grateful to the reviewers for their insightful comments on our paper. We have been able to incorporate changes to reflect most of the suggestions provided. We have highlighted the changes within the manuscript. Here is a point-by-point response to the reviewers’ comments and concerns
Comment 1. Support recommendations with evidence: Strengthen the proposed strategies for enhancing eHealth capacity building by citing relevant evidence or existing literature to demonstrate their effectiveness and feasibility.
The above comment suggested authors to compare the outcomes with previous studies from literature. To support the outcomes and directions in present study
Response: Thank you for pointing this out. We have included some revision to include your suggestions. You can find the explanation in the "Discussion section," and on the last paragraph.
Comment 2: Also, The Piecharts should be on white background and should be organized well. The plots are not satisfactory.
Response: Thank you for pointing this out. We have included the comments you raised and have redrawn the pie charts. See pages 7 and 11.
We look forward to hearing from you
Your sincerely,
Flora